# Deep convolutional neural networks for regular texture recognition

Ni Liu[1], Mitchell Rogers[2], Hua Cui[1], Weiyu Liu[3], Xizhi Li[4] and Patrice Delmas[2]

[1] School of Information Engineering, Chang'an University, Xi'an, ShaanXi Province, China
[2] Department of Computer Science, The University of Auckland, Auckland, New Zealand
[3] School of Electronics and Control Engineering, Chang'an University, Xi'an, China
[4] Henan Highway Development Co. Ltd. Anxin Branch, Xinxiang City, Henan Province, China

## ABSTRACT

Regular textures are frequently found in man-made environments and some biological and physical images. There are a wide range of applications for recognizing and locating regular textures. In this work, we used deep convolutional neural networks (CNNs) as a general method for modelling and classifying regular and irregular textures. We created a new regular texture database and investigated two sets of deep CNNs-based methods for regular and irregular texture classification. First, the classic CNN models (*e.g.* inception, residual network, *etc.*) were used in a standard way. These two-class CNN classifiers were trained by fine-tuning networks using our new regular texture database. Next, we transformed the trained filter features of the last convolutional layer into a vector representation using Fisher Vector pooling (FV). Such representations can be efficiently used for a wide range of machine learning tasks such as classification or clustering, thus more transferable from one domain to another. Our experiments show that the standard CNNs attained sufficient accuracy for regular texture recognition tasks. The Fisher representations combined with support vector machine (SVM) also showed high performance for regular and irregular texture classification. We also find CNNs performs sub-optimally for long-range patterns, despite the fact that their fully-connected layers pool local features into a global image representation.

## INTRODUCTION

Textures are composed of atomic units called textons. The layout of textons can be regular or irregular (*Hettiarachchi, Peters & Bruce, 2014*). Regular textures exhibit strongly periodic or quasi-periodic behavior, often found on building surfaces, bricks, floor tiles, fences, vegetation and crop fields. A period describes the distance between repeating textons. For images, we consider the spatial distance. In contrast to regular textures, irregular textures have "noisy" or "stochastic" patterns with random intensity variations. In practice, strong periodicity rarely occurs in nature, these textures are usually quasi-periodic, which means patterns recur, but the periodicity has random components (*Hettiarachchi, Peters & Bruce, 2014*; *Liu et al., 2015*).



Corresponding author
Ni Liu, niliu@chd.edu.cn

The layout of regular textures are an important property that allows an object to be correctly identified. The recognition and segmentation of regular textures could be used across many domains such as computer vision, computer graphics, and medical imaging (*Cai & Baciu, 2011*; *Sun, Kingdom & Baker, 2019*). Applications of regular texture detection include: analysis and quantification of building patterns for urban planning (*Yu et al., 2017*), terrain segmentation (*Aksoy, Yalniz & Tasdemir, 2012*), texture replacement in images (*Liu, Lin & Hays, 2004*; *Hettiarachchi, Peters & Bruce, 2014*), texture synthesis (*Lin et al., 2006*), quantification of drosophila eye surface regularity (*Diez-Hermano et al., 2020*), and pattern segmentation in woven fabric (*Cai & Baciu, 2011*).

The analysis of repetitive patterns is a long standing problem in texture analysis (*Yu et al., 2017*). A pioneering work by Leung and Malik in the 90's used a spatial tracking approach to find repeating elements of images (*Leung & Malik, 1996*). First, they detected windows of possible candidates for textons. Then, neighboring regions of each candidate were searched for similar structures. Regions with similar repeating elements were grouped together, and individual elements were marked. The main drawback of this method is that it did not consider complex textons, periods or texton regularity. In *Schaffalitzky & Zisserman (1998)*, they proposed a grouping algorithm based on local affine transformations. However, their method was sensitive to structural distortions. More recently, the texton grouping problem was reformulated as a lattice detection problem (*Hays et al., 2006*; *Park et al., 2009*). In *Hays et al. (2006)* the solution is found by iteratively proposing textons and assigning neighbors to the textons. In *Park et al. (2009)* a Markov Random Field (MRF) with a Mean-Shift Belief Propagation method was used. Other approaches to texton grouping include optimization of shape alignment (*Cai & Baciu, 2011*), structural regularity using symmetry groups (*Liu et al., 2008*), projection profiles (*Aksoy, Yalniz & Tasdemir, 2012*) and frequency filtering (*Hettiarachchi, Peters & Bruce, 2014*; *Sun et al., 2021*).

As opposed to developing a specialized method for specific scenes and applications, our objective is to develop a general method for recognizing texture regularity, that can be easily applied to regular texture related areas only with light fine-tuning needed. According to the level of regularity, we classify textures into two categories, regular or irregular. This configuration allows us to seek standard representations of regular textures by learning their common characteristics with an easy collection of a large number of samples. Then the developed method can be fine-tuned into specific areas with only a few labeled images needed. We envision that our work could be used to enhance image retrieval, object recognition and 3D structure identification tasks (*Leung & Malik, 1996*). For this challenging task, we therefore decided to employ the power of deep convolutional neural networks (CNNs).

CNNs have been shown to outperform traditional approaches for image classification tasks. This prompted the quick adoption of deep learning methods in many image classification fields such medical (*Litjens et al., 2017*; *Wang et al., 2021*; *Singh, Sengupta & Lakshminarayanan, 2020*), aerial (*Mnih & Hinton, 2012*; *Petrovska et al., 2020*), vehicle recognition (*Adu-Gyamfi et al., 2017*), traffic congestion recognition (*Cui et al., 2020*), gait

recognition (*Sepas-Moghaddam & Etemad, 2021*), fruit recognition (*Murean & Oltean, 2018*; *Saedi & Khosravi, 2020*) as well as for general images (*Krizhevsky, Sutskever & Hinton, 2012*; *Simonyan & Zisserman, 2014*; *He et al., 2016*; *Szegedy, Toshev & Erhan, 2013*). Unlike the first perceptrons (*Rosenblatt, 1958*) which have a single layer of neurons, deep learning models consist of multiple layers of inter-connected neurons (*Widrow & Lehr, 1990*) with back-propagation (*Rumelhart, Hinton & Williams, 1988*). CNNs can learn increasingly complex representations of objects by exploiting the compositional nature of images. High-level features like faces are composed of lower-level features such as edges and lines (*Llamas et al., 2017*). In CNNs, the lower layers correspond to corners, edges and color conjunctions, while deeper layers correspond to more complex high-level features such as faces, text, wheels and flowers (*Hossain & Serikawa, 2013a*). CNN models have achieved state-of-the-art performance for image recognition with architectures such as LeNet-5 (*LeCun et al., 1998*), AlexNet (*Krizhevsky, Sutskever & Hinton, 2012*), VGGNet (*Simonyan & Zisserman, 2014*), and more recently Inception network (GoogLeNet) (*Szegedy et al., 2015*; *Szegedy et al., 2016b*), and Residual network (ResNet) (*He et al., 2016*). Most of these architectures feature deeper layers, such as Inception network which has 22 layers.

In this paper, we investigated two sets of CNN based methods for regular texture modelling and classification. First, the classic CNN models were used in a standard way. The output layer was set to two categories while other layers maintained their original architectures. These two-class classifiers were trained using our newly created regular texture. The following state-of-the-art CNN models were tested: Inception network, Residual network and Inception-ResNet-v2 (*Szegedy et al., 2016a*). Second, we applied Fisher Vector pooling (FV) to the learned features (filter responses) of the above trained CNNs (*Perronnin & Dance, 2007*). The generated vector representations, combined with classifiers such as SVM, can be efficiently used for a wide range of machine learning classification or segmentation tasks, and as such are more transferable from one domain to another. The motivation for this method stems from the state-of-the-art texture analysis method FV-CNN (*Cimpoi et al., 2015*), proposed by the Visual Geometry Group of Oxford University. However, they only used pre-trained VGG features on ImageNet ILSVRC data, which may not be optimal for modelling texture regularity.

The main contributions of our paper are: (1) A newly created regular texture database, which is publicly available. This database complements well the state-of-the-art Describable Textures Dataset (DTD) from the Visual Geometry Group (*Cimpoi et al., 2014*). Their database is comprised of visual perception of textures in features such as dotted, bumpy, cracked, striped and line-like and so on. Their dataset mainly focused on textons (texture elements) and did not consider the global configuration of textons and the degrees of regularity in textures. (2) A generalized regular texture modelling and recognition framework. The trained deep CNN models and further generated Fisher representations were robust to different texture layouts, pattern complexity, texton variability and viewing angles, and is transferable from one domain to another. Our experiments showed that both methods reached remarkable accuracy, with a best performance of 98% for general regular texture classification.

## METHODS

In this work, our main goal was to classify texture images by assigning them to a specific label. We used Convolutional Neural Networks to learn from a set of labelled training data. Then, the model was used to predict class labels for a test set which has not been seen during training. Here, we consider two texture categories, namely, regular or irregular patterns. The images were manually labelled by volunteers using a consensus approach.

### Regular texture database

A considerable number of images are needed in order to train CNNs. Several image databases have been developed for generic texture analysis including the natural texture image database, texture of materials database, and dynamic texture database (*Hossain & Serikawa, 2013b*). However, they are mainly texture image banks and contain limited regular textures. Only a small number of databases, CMU Near-Regular Texture (NRT) database (*Liu, Lin & Hays, 2004*) and PSU Near-Regular texture database (PSU-NRTDB, 2005, http://vivid.cse.psu.edu/) provide near-regular textures. However, these databases are small in size and the data is mostly unlabelled. A related dataset is the Describable Texture Dataset (DTD) (*Cimpoi et al., 2014*), which categorizes textures by adjectives *e.g.*, dotted, cracked, or wrinkled. This database focuses more on textons rather than the global layout of textons. From these categories, we identified a subset of classes "grid, banded, chequered, grooved, meshed" that can be used to describe regular textures.

We created our own texture database which focused on regular elements. Our database consists of two categories: regular textures and irregular textures. We manually curated this database by selecting relevant textures from existing online libraries:

- PSU near-regular texture (PSU-NRTDB, 2005, http://vivid.cse.psu.edu/)
- Texturelib (*Chugai, 2019*)
- Brodatz dataset (*Randen, 2012*)
- Columbia-Utrecht Reflectance and Texture (CUReT) (*Dana et al., 1999*)
- Describable Texture Dataset (DTD) (*Cimpoi et al., 2014*)
- Kylberg database (*Sarafraz, 2011*)
- Texture Library (*Francv, 2010*)
- Sketchuptexture (*SketchupTextureClub, 2010*)

To supplement our database, we also obtained images from Flickr (*MidCenturyStyles, 2015*) and Google images (*Bayless, 2019*). The number of images based on source location and image class are listed in Table 1. Our regular database has a total of 1,230 regular textures and 1,230 irregular textures, including images of different sizes.

As color information is irrelevant to the layout of textons, in our experiments, all images are first transformed to grey-scale. Images are then expanded into three channels by repeating the gray-scale data. Figure 1 shows examples of the two image categories in our database.

**Table 1 The number of regular and irregular textures selected from online resources.**

| Database 1 | # Regular | # Irregular |
| --- | --- | --- |
| PSUN | 335 | 98 |
| Texturelib | 363 | 588 |
| Brodatz | 36 | 46 |
| CUReT | 2 | 0 |
| DTD | 216 | 459 |
| Kylberg | 7 | 19 |
| Texture Library | 16 | 20 |
| Sketchup texture | 14 | 0 |
| Google images | 40 | 0 |
| Flickr | 201 | 0 |
| Total number | 1,230 | 1,230 |

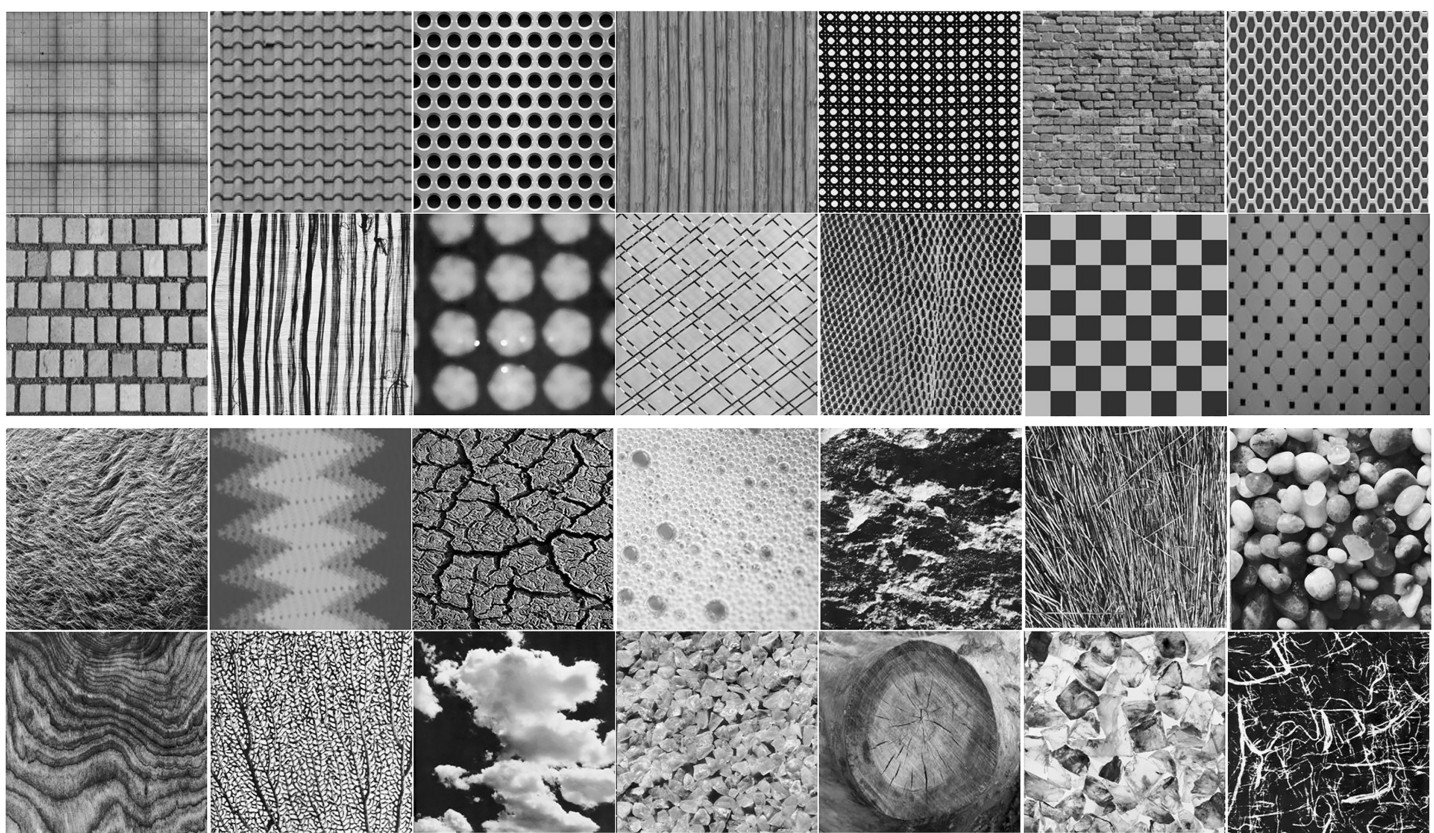

**Figure 1 Examples of regular (top two rows) and irregular textures (bottom two rows) from our database.** Our database has a total of 1,230 regular textures and 1,230 irregular textures.

## Deep convolutional neural networks

CNNs are used commonly for image classification. The CNN architecture is formed by several intermediate layers, which are typically convolutional layers and pooling layers.

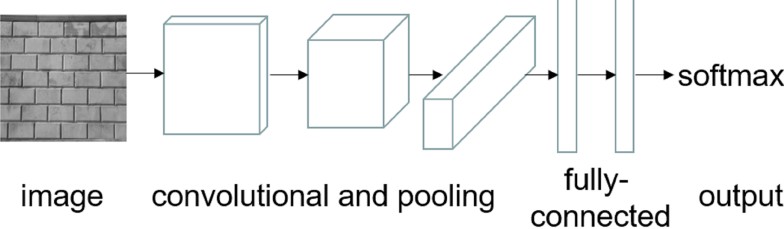

**Figure 2 A typical convolutional neural network.**

Followed by a fully-connected layer which performs the final classification using the output from the previous layers. Figure 2 shows an example of a simple CNN architecture.

The layers of a CNN are described in detail below.

Convolution layer: The convolutional layer is the core building block of a CNN. A small filter (usually 3 × 3) moves over the image, generating a map of activation called a features map. This makes certain characteristics become more dominant in the output image. For example, edges can be detected by filters that highlight the gradient in a particular direction. The output of each layer can be formulated as:

$$\mathbf{a}^l = \sigma(\omega^l * \mathbf{a}^{l-1} + \mathbf{b}^l) \tag{1}$$

where $l$ represents the $l$th layer, $*$ is a convolution operation (filter), $\boldsymbol{\omega}^l$ is the weight matrix, $\mathbf{b}^l$ is the vector (bias) and $\sigma$ is the nonlinear activation function.

Activation layer: After the convolution, non-linear activation functions are applied to the features maps. The most common used is the ReLU (Rectified Linear Unit): $f(x) = \max(0, x)$. This function propagates the gradient efficiently and alleviates the problem of vanishing gradient when there are many layers.

Pooling: The pooling or down-sampling layer is responsible for reducing the size of the activation maps. Grouping operations (max-pooling) find the maximum value of a sample window and pass this value as a summary of that area. As a result, the size of the data is reduced by a factor equal to the size of the sample window. In general, it reduces the computational power requirements progressively through the network. It also reinforces invariance properties to small changes such as feature position and image distortion.

Fully-connected layer (FCL): The objective of a fully-connected layer is to take the results of the convolution/pooling process and use them to classify the image. Adding a fully-connected layer is beneficial when learning combinations of non-linear features.

Training a neural network consists of minimizing a global error function. The error function calculates the error between the output class and the true class, and averages this error over all input images. The training steps are: (i) calculate the output class using feedforward; (ii) calculate the error; (iii) back-propagate the gradients and update the weights.

Stochastic gradient descent iteratively minimizes the errors by updating the network weights in the opposite direction to the gradient of the cost function (*Rumelhart, Hinton & Williams, 1988*). The frequency of the error updates can be done per batch, per sample

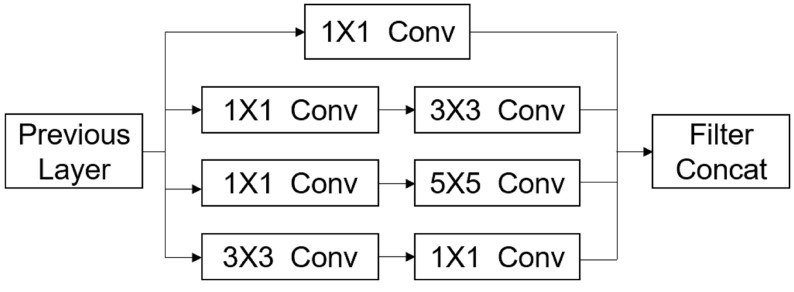

**Figure 3 Inception module.**               

or anywhere in between. Typically mini-batches give the best balance between computational time and accuracy.

For this method, there are many optimizations; learning rate, weight decay (a regularization term, which penalizes changes in the weights and prevents them from being too large), and momentum (the average of the previous gradients which reduces oscillations that cause local minima, thus accelerating convergence).

The following equation is used to update the values of the network weights $w_i$:

$$v_{i+1} := r \times v_i - \lambda \times \varepsilon \times w_i - \varepsilon \times \left\langle \frac{\partial L}{\partial \omega}|_{\omega_i} \right\rangle_{\mathbf{D}_i} \tag{2}$$

$$\omega_{i+1} := \omega_i + v_{i+1} \tag{3}$$

where $i$ is the iteration index, $\varepsilon$ is the learning rate, $\lambda$ is the weight decay, $v$ is the momentum variable, $r$ is the momentum weight and $\left\langle \frac{\partial L}{\partial \omega}|_{\omega_i} \right\rangle_{\mathbf{D}_i}$ is the average error over the $i$th batch $\mathbf{D}$ of the derivative of the objective function with respect to $\omega$, evaluated at $w_i$. The cross-entropy loss function was: $L(\omega) = -\sum_j \sum_c y_{jc} \log f_c(g_j)$, where $y_{jc}$ denotes the label for an image with indexed $j$ and class $c$; $f_c(g_j)$ is the prediction probability of class $c$ for image $g$.

## Regular textures classification

We investigated two sets of Deep CNNs based methods for modelling regular textures and classifying regular and irregular textures.

*Standard Deep CNNs* First, we used classic CNNs in a standard way. The output layer was set to the number of texture categories while other layers maintained their original architectures. These two-classes classifiers were trained using our newly created regular texture database, by fine-tuning ImageNet pre-trained networks. Further details for the three state-of-the-art CNNs are introduced as follows.

The Inception-v3 network (*Szegedy et al., 2016b*) uses Google's Inception architecture (*Szegedy et al., 2015*) for image recognition. The "Inception" modules concatenate filters of different sizes and dimensions into a single new filter. They use a combination of a small kernel (1 × 1, 3 × 3, and 5 × 5 convolutions) and a few convolutional filters with large kernel size (Fig. 3). Convolutions of different sizes were used to capture details at various scales. Another feature of the module is a bottleneck layer of 1 × 1 convolutions. This is designed to reduce the computational complexity.

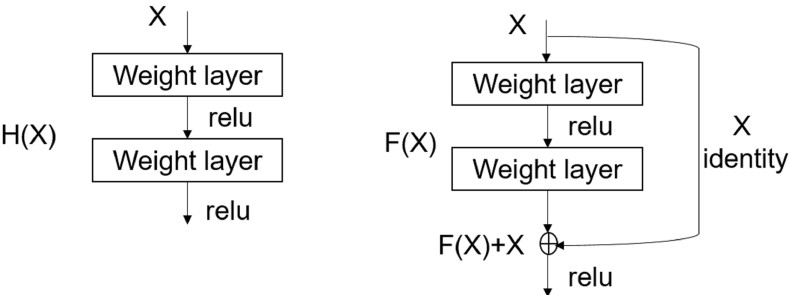

**Figure 4 Normal convolutional neural network (left); shortcut connections of ResNet architecture (right) (*He et al., 2016*).**

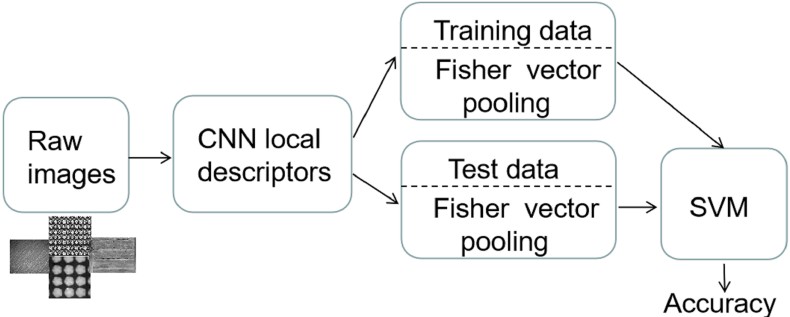

**Figure 5 Framework for Fisher vector representations of CNN features, combined with SVM for texture classification.**

The residual network (*He et al., 2016*) introduces an identity function between layers. Conventional networks learn underlying nonlinear mappings $y = \mathcal{H}(x)$ of stacked layers (Fig. 4 (left)), with $x$ denoting the inputs to the first of these layers. In residual networks, a new nonlinear function $y = \mathcal{F}(x) + x$, here $\mathcal{F}(x)$ is called the residual (Fig. 4 (right)). The Residual mapping avoids the vanishing gradient problem, so the depth of the neural network can be increased without increasing the number of parameters to optimize.

The Inception-ResNet-v2 network (*Szegedy et al., 2017*) is considered as state-of-the-art for ImageNet recognition. Residual connections include shortcuts in models that allow researchers to train even deeper networks to achieve a higher performance. This has also allowed Inception blocks to be further simplified.

*Fisher Vector Representations* The above standard CNN-based methods can be interpreted as extracting local convolutional features and pooling them in a global image representation by the Fully-Connected (FC) layers. In *Cimpoi et al. (2015)*, FC pooling was replaced by Fisher Vector pooling (FV) in the pre-trained VGG network (on ImageNet ILSVRC data) for general texture representation. This allowed for state-of-the-art performance across various texture databases and an efficient method benefit in transferring features from one domain to another. Motivated by their work, we further applied Fisher Vector pooling to the filter features of the last convolutional layer of the above learned standard CNNs. Vector representations generated in this way are

quite suitable for tasks such as classification with Support Vector Machine (SVM). They proved more transferable from one domain to another and can be applied out-of-the-box without extensive fine-tuning, with only a few labeled images needed to train SVM.

Our framework (see Fig. 5) first extracts deep convolutional features from the last convolutional layers of CNNs trained on our regular texture dataset. Fisher Vector pooling (FV) maps a sequence $\mathscr{F} = (\mathbf{f}_1, ..., \mathbf{f}_n), \mathbf{f}_i \in \mathbb{R}^D$ of local convolutional descriptors to a feature vector, where $D$ is the dimensionality of descriptors. Local descriptors are first assigned to elements in a visual dictionary, constructed by a $K$ modes Gaussian Mixture Model (GMM) $(\pi_k, \mu_k, \Sigma_k), k = 1, ..., K$, where $\pi_k \in \mathbb{R}$ is the prior probability of the component, $\mu_k \in \mathbb{R}^D$ the Gaussian mean and $\Sigma_k \in \mathbb{R}^{D \times D}$ the Gaussian covariance. The assignments $\eta(\mathbf{f}_i)$ are given by the posterior probability of each GMM component. Rather than storing visual word occurrences only, these representations store a statistics of the difference between dictionary elements and pooled local features. The FV descriptor encoder $\eta\mathrm{FV}(\mathbf{f}_i)$ includes both first order $\Sigma_k^{-\frac{1}{2}}(\mathbf{f}_i - \mu_k)$ and second order statistics $\Sigma_k^{-\frac{1}{2}}(\mathbf{f}_i - \mu_k) \odot (\mathbf{f}_i - \mu_k) - \mathbf{1}$ (see (*Perronnin & Dance, 2007*) for more details). The dimensionality of the Fisher representation is $2KD$. Learning uses a standard non-linear SVM solver. We denote this set of methods as FV-CNNreg.

## Experimental setup

In our experiments, we evaluate the performance of standard CNNs and FV-CNNreg for distinguishing regular and irregular textures. We set a 7:1:2 ratio in our database, resulting in 860, 120, and 250 texture images for training, validation and testing for each category. We used 10-fold cross-validation and the averages of 10 splits for performance comparison. Input images were pre-processed to standardized size and transformed to grey-scale. All the experiments involving CNNs training were implemented on the Keras platform (*Chollet, 2015*). The VLFeat library in MATLAB was used for the computation of FV and SVM solvers.

The training of these networks in Keras requires the adjustment of hyperparameters including learning rate, weight decay, momentum *etc*. The initial learning rate is often the most important hyperparameter. Its value is usually less than 1 and greater than 1e−6. Usually, 0.01 is used as a typical value. Keras provides the Stochastic Gradient Descent (SGD) class that implements the stochastic gradient descent optimizer with a learning rate decay and momentum. The learning rate is calculated at the end of each mini-batch as follows (*Brownlee, 2020*):

$$\varepsilon_i = \varepsilon_0 \times \frac{1}{(1 + \lambda \times i)} \tag{4}$$

where $\varepsilon_i$ is the learning rate for the current iteration $i$; $\varepsilon_0$ is the initial learning rate specified as an argument to SGD and $\lambda$ is the decay rate which is greater than zero. We investigated the effect of a set of different values of the learning rate decay for CNNs in the Results section. We also compared the effect of using data augmentation features such as the shift, flip and rotation from Keras.

**Table 2 Comparison of accuracy (%) for standard CNN models for 10-fold cross-validation experiments.**

| Folds | ResNet-50 | InceptionV3 | Inception-ResNet-v2 |
|---|---|---|---|
| 1 | 94.00 | 96.80 | 97.40 |
| 2 | 91.00 | 96.00 | 97.00 |
| 3 | 91.00 | 96.80 | 98.20 |
| 4 | 93.80 | 96.80 | 98.80 |
| 5 | 93.10 | 96.20 | 96.60 |
| 6 | 95.60 | 97.60 | 98.40 |
| 7 | 96.20 | 96.20 | 97.60 |
| 8 | 95.20 | 96.40 | 97.80 |
| 9 | 97.40 | 97.60 | 97.40 |
| 10 | 96.40 | 97.80 | 98.60 |
| Mean | 94.37 | 96.82 | 97.78 |
| Std. dev. | 2.20 | 0.65 | 0.71 |

**Table 3 Hyperparameters used in fine-tuning the three standard CNNs.**

| Model | Initial learning rate | Learning rate decay | Batch size | Epoch |
|---|---|---|---|---|
| Inception-ResNet-v2 | 0.01 | 0.01 | 32 | 200 |
| InceptionV3 | 0.01 | 0.1 | 32 | 200 |
| ResNet-50 | 0.01 | 10 | 32 | 200 |

**Table 4 The number of images misclassified by Inception-ResNet-v2.** Type1 denotes the number of regular images classified as irregular. Type2 denotes the number of irregular images classified as regular. The test data has 250 regular and 250 irregular textures.

| Folds | 1 | 2 | 3 | 4 | 5 | 6 | 7 | 8 | 9 | 10 |
|---|---|---|---|---|---|---|---|---|---|---|
| Type1 No. | 8 | 4 | 2 | 2 | 7 | 5 | 3 | 2 | 3 | 4 |
| Type2 No. | 5 | 11 | 7 | 4 | 10 | 3 | 9 | 9 | 10 | 3 |

## RESULTS

In the first experiment, we evaluated the performance of standard CNN models. We report the classification accuracy in Table 2 for three state-of-the-art classical CNN models, InceptionV3, ResNet-50 and Inception-ResNet-v2. We also evaluated several combinations of the parameters of the neural networks to be tuned. The hyperparameters finally used for offering this best results are presented in Table 3. The initial learning rate $\varepsilon_0 = 0.01$ and the momentum $v = 0.9$ are typical choices (*Brownlee, 2020*). The confusion matrix is shown in Table 4.

Inception-ResNet-v2 appears to be the most robust model as seen in Table 2. This is mainly attributed to the beneficial combination of the Inception architecture with residual connections creating a large-scale architecture with more layers and parameters than other CNNs. Figure 6 shows the improvement of the accuracy during the training phase of the Inception-ResNet-v2 network using the validation images of our database. It also

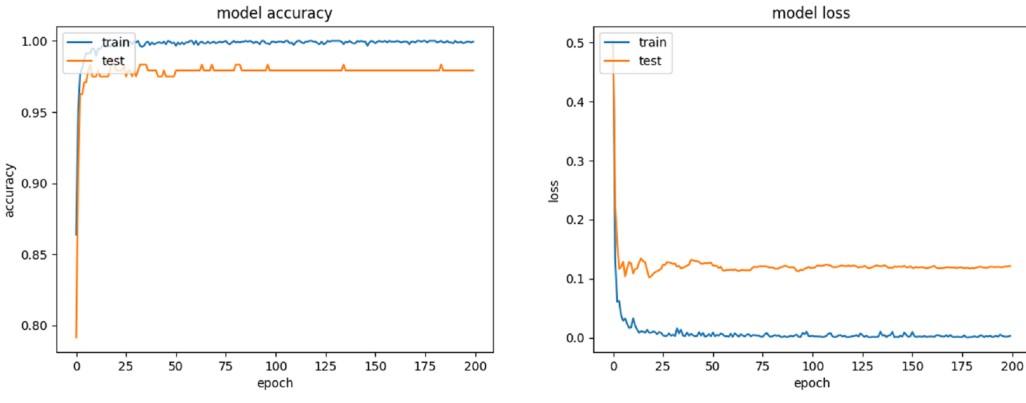

**Figure 6 Training and validation accuracy of Inception-ResNet-v2 and associated cost during training.**

demonstrated how the associated cost reduced during training in one experiment. The number of images misclassified by Inception-ResNet-v2 is shown in Table 4.

To better understand the performance benefits of Inception-ResNet-v2, we analyzed correctly classified images (with class probabilities $p \geq 0.50$, see Fig. 7), and misclassified images (with class probabilities $p < 0.50$, see Fig. 8). Images are numbered in row-major order. The predicted class probabilities are shown below each image. We observed that the CNN correctly recognized most regular patterns: wallpapers (images 2–7), store shells (images 8 and 9), architectural surfaces (images 10–13), floor patterns (images 15–18), stone wall (image 19–21), fabric patterns (images 22–25). We can see that these patterns vary in shape. These easy cases include dotted, striped, lattice, chequered and honeycombed textons. Regular patterns with different viewing angles are also correctly recognized (images 10–14). We observed that near-regular textures usually have lower probabilities.

As shown in Figs. 1 and 7, our regular texture database contains many heterogeneous images from a variety of scenes. The high accuracy achieved by Inception-ResNet-v2 suggests CNNs are highly generalizable. This also suggests that the database we built is reasonable for regular texture recognition tasks. We believe CNNs have the potential to be a powerful tool for building general methods for recognizing regular textures.

The wrongly classified examples are all shown in Fig. 8. These images have low predicted probabilities ($p < 0.50$). It is important to take a close examination of these images, in order to deeply understand the performance of the trained CNN on recognizing regular textures. The first wallpaper textons have a regular layout, but very complex patterns. This can be seen as an outlier as most patterns are relatively simple. The stonewall (image 2) is a borderline case, with an accuracy of 0.35. We found that images with light edges are not easily recognizable. The planks (image 3) is misclassified, with an accuracy of 0.01. We can see that the each pile has a noisy surface (stochastic), and only the gaps between piles has a regular pattern. This shows the CNNs donot use long-range regular structures. Similarly for other misclassified images. They both show high levels of noise within the larger regular patterns.

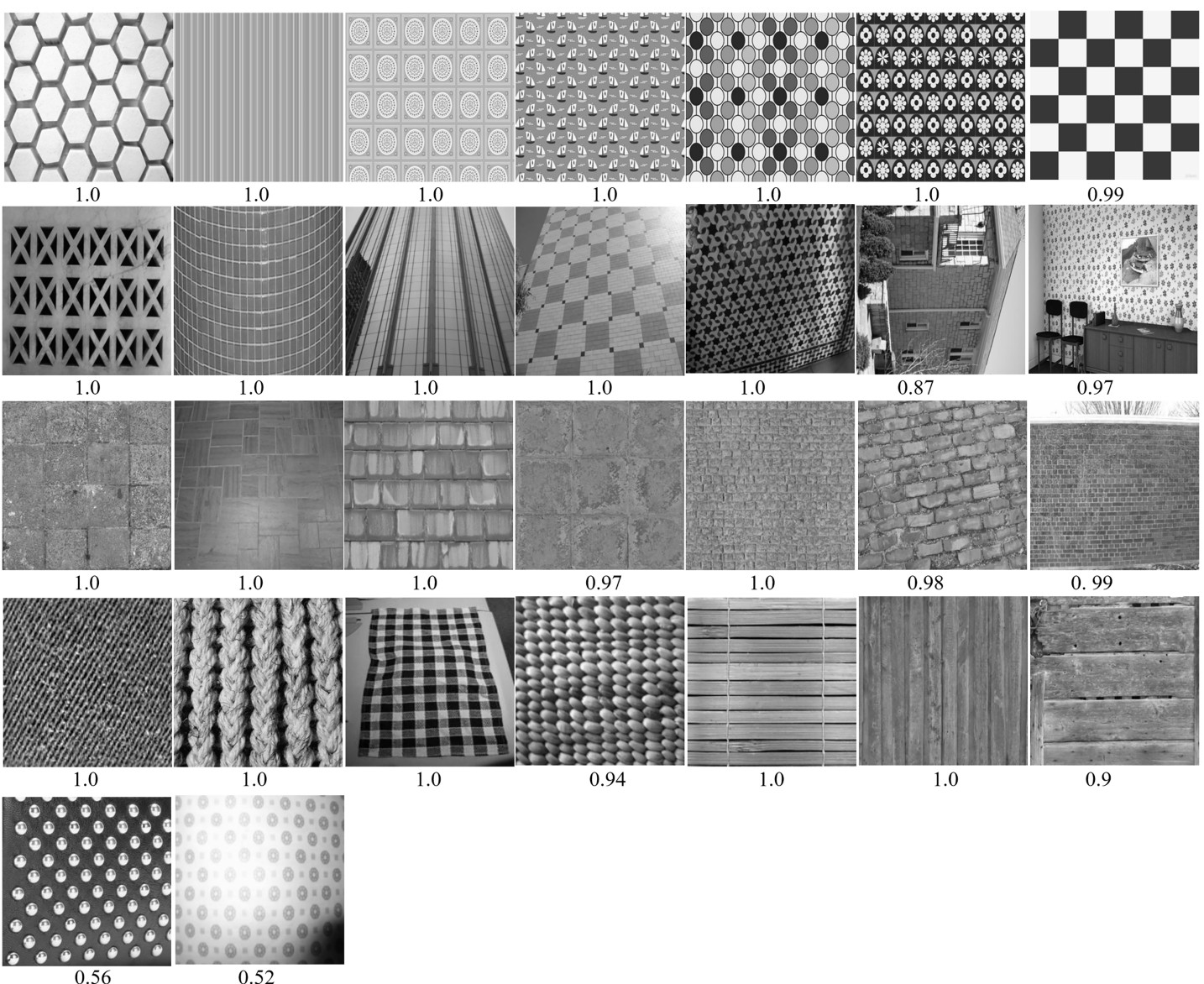

**Figure 7 For regular textures, examples of correctly classified by Inception-ResNet-v2.** Images are numbered from 1 to 30 in row-major order. Images were scaled to 256 × 256 for visualization. Predicted class probabilities are listed under each image.

Figure 9 shows the results for the irregular textures. The wrongly classified examples are shown in the bottom row. For images 22, 23, 24, 25, 26 and 27, potential causes are the examples possess a considerable amount of sharp edges, and no stochastic random texture contained in those irregular textures. This misled the CNN to only consider local structures, which made it wrongly classify them as regular textures. Possible solutions include adding similar samples in the training irregular images. Texture synthesis is a good way to generate irregular images by breaking regularity of existed regular textures. This provide a future direction to improve our irregular samples.

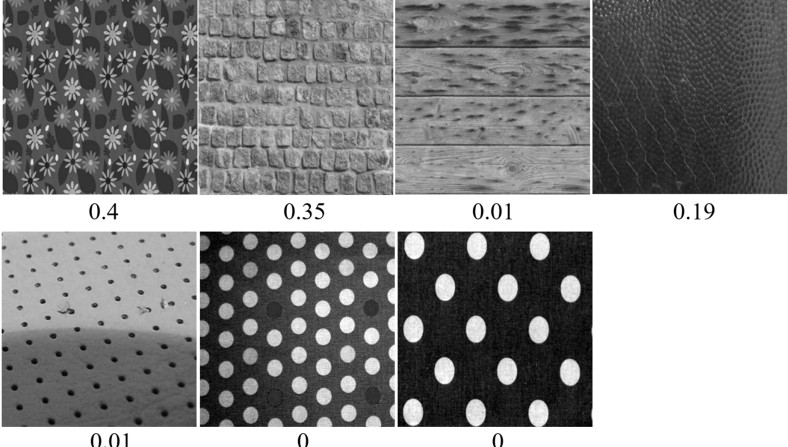

**Figure 8** **For regular textures, examples of wrongly classified images by Inception-ResNet-v2.** Images are numbered from 1 to 7 in row-major order. The predicted class probabilities are shown below each image.                                                       

In summary, standard CNNs such as Inception-ResNet-v2 achieved a high accuracy of 97.78% for classifying regular textures and irregular textures. The CNN correctly recognized most regular patterns, but had lower accuracy on borderline cases (near-regular textures) and hybrid textures (*e.g.* tiles of noisy textures).

In the second experiment, we investigated the tuning of learning rate decay for the three standard CNN models using fold 1 data. We set the batch size and epoch as 32 and 200, and evaluated a set of the learning rate decays for the three CNN models. With the chosen model configuration, the results suggested a moderate learning rate decay of 0.01, 0.1 and 10 resulted in good performance for Inception-ResNet-v2, InceptionV3 and ResNet-50, respectively. Figure 10 shows oscillations in behavior for the large learning rate of 0.1 for Inception-ResNet-v2, and the inability of the model to learn with the small learning rate of 0.01 for InceptionV3 and 0.1 for ResNet-50.

In the above experiments, we used data argumentation provided by Keras including horizontal and vertical flip, rotation range of 10, width and height shift range of 0.05. Figure 11 compares the training process between using the data argumentation and not. It can be observed that when the data argumentation was used (left), the model took much less time to achieve convergence and had higher training accuracy.

In the third experiment, we evaluated the performance of the further generated Fisher representations of the above trained CNNs for classifying regular and irregular textures. Specifically, we transformed the filter features of the last convolutional layer of the trained Inception-ResNet-v2 into Fisher vector representation. A standard SVM solver was used as classifier. We denoted this method as FV-CNNreg-IncepResv2, as introduced in the Methods section. The last layer of convolutional filters of Inception-ResNet-v2 outputs features of dimension $8 \times 8 \times 1,536$. Dimensionality reduction (PCA) was further applied to the CNN local descriptors before FV pooling, reducing the output features dimensions to 512 and providing the best performance in our test. Gaussian Mixture Model (GMM) was set to $K = 64$ modes. The traditional SIFT features were also compared.

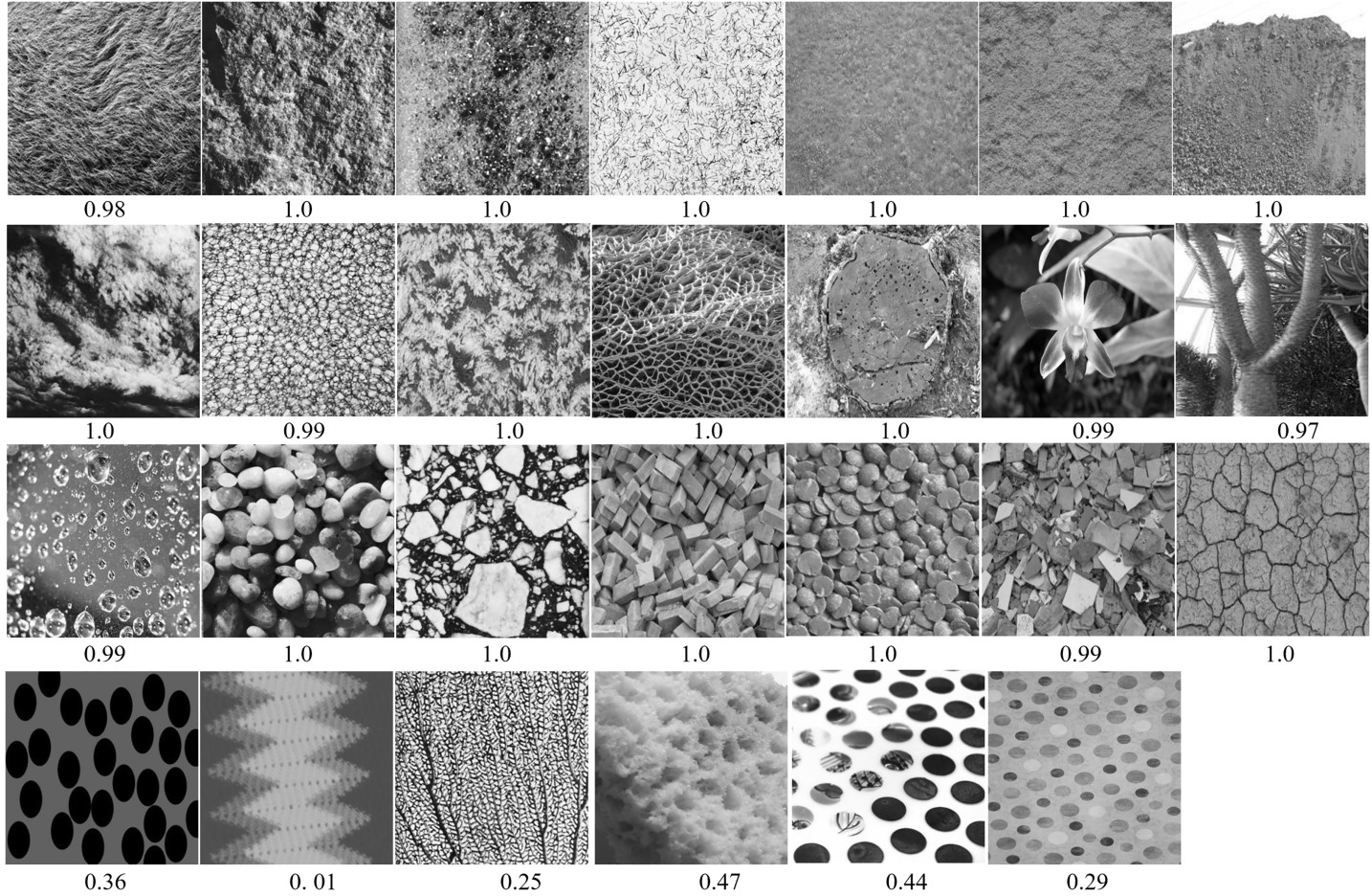

**Figure 9 For irregular textures, examples of correctly classified (top three rows) and all wrongly classified images by Inception-ResNet-v2.** Images are numbered from 1 to 27 in row-major order. The predicted class probabilities are shown below each image.

Three kernels were exploited for the SVM classifier, including linear kernel, Hellinger's kernel (short for Hell) and $\chi^2$ kernel (short for Chi2) (*Maji, Berg & Malik, 2008*; *Schölkopf, Smola & Bach, 2002*). The SVM solvers was implemented using the VLFeat library. The hyperparameters were set as follows. Let $n$ denote the total number of training samples (both positive and negative), regularization coefficient was $\frac{1}{n}$, stopping criterion epsilon 0.001 and maximum iterations $n \times 200$. Results are shown in Table 5.

Fisher vector representation, FV-CNNregIncepresV2, gave a good result of 97.71%. This proves the usefulness of transforming deep CNNs feature into Fisher vector representations. In comparison, Fisher vector representation of traditional SIFT features has a worse accuracy (90.76%) than all the CNN based methods.

## DISCUSSIONS

Regularity is an important high-level characteristic of textures toward perception understanding. The main challenge of modelling the regularity of textures is that it is not obvious what kind of features should be extracted. We utilize the deep convolutional

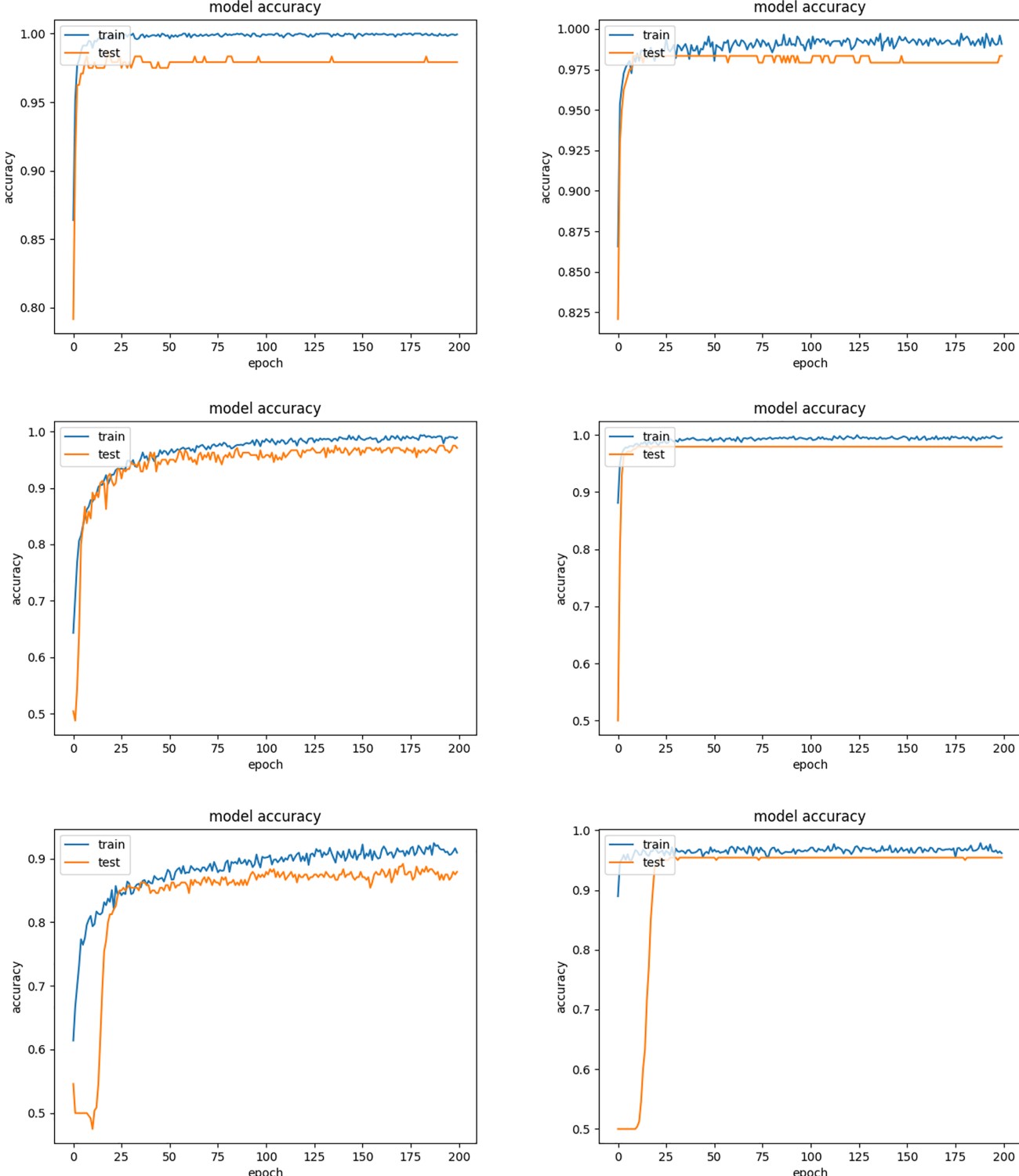

**Figure 10 Train and validation accuracy for different learning rate decays ε.** Top: Inception-ResNet-v2 (ε = 0.01 *vs.* ε = 0.1). Middle: InceptionV3 (ε = 0.01 *vs.* ε = 0.1). Bottom: ResNet-50 (ε = 0.1 *vs.* ε = 10).

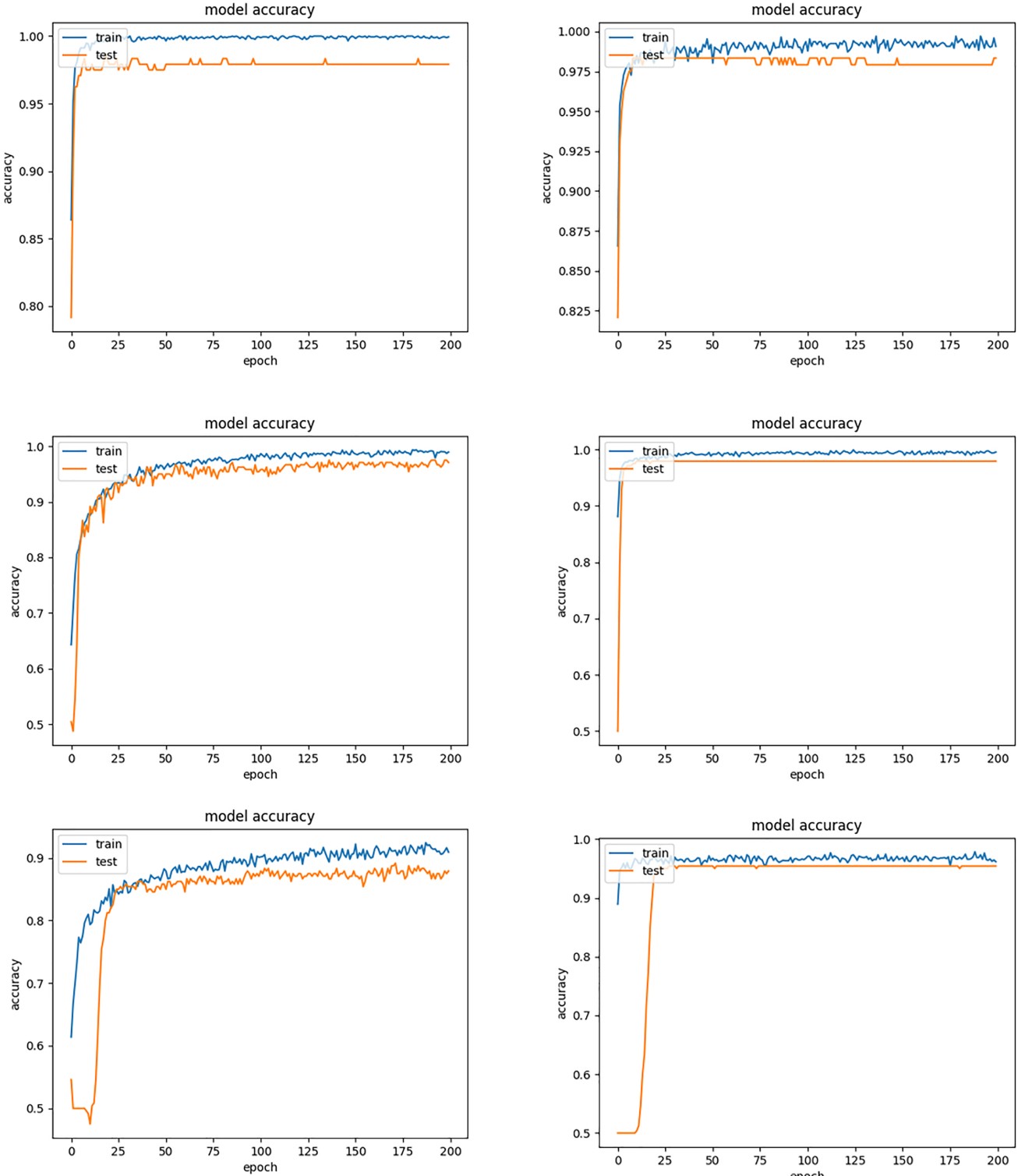

**Figure 11 Training accuracies of Inception-ResNet-v2 between using data argumentation (left) and not (right).**

**Table 5 Comparison of accuracy (%) for Fisher vector representations for 10-fold cross-validation experiments.** FV-CNNreg-IncepResv2 applies the Fisher pooling to the last convolutional layer of the trained Inception-ResNet-v2. FV-SIFT denotes the Fisher pooling of SIFT features. Note that linear means linear kernel, Hell denotes the Hellinger's kernel, and Chi2 represents $\chi^2$ kernel.

| Model/folds | FV-SIFT | | | FV-CNNreg-IncepResv2 | | |
|---|---|---|---|---|---|---|
| Kernel | Linear | Hell | Chi2 | Linear | Hell | Chi2 |
| 1 | 89.60 | 89.40 | 89.40 | 98.00 | 98.00 | 98.00 |
| 2 | 90.4 | 90.00 | 89.20 | 97.20 | 96.40 | 96.80 |
| 3 | 90.00 | 87.60 | 90.40 | 97.80 | 98.40 | 98.40 |
| 4 | 91.20 | 90.20 | 90.00 | 97.80 | 98.00 | 98.20 |
| 5 | 92.40 | 91.60 | 91.60 | 96.20 | 96.60 | 96.60 |
| 6 | 91.00 | 90.00 | 90.00 | 98.20 | 97.80 | 97.80 |
| 7 | 90.00 | 89.20 | 92.20 | 97.40 | 97.40 | 97.80 |
| 8 | 91.20 | 89.40 | 91.00 | 97.40 | 97.40 | 97.60 |
| 9 | 91.20 | 92.30 | 91.00 | 98.00 | 98.60 | 97.80 |
| 10 | 90.60 | 88.60 | 90.20 | 98.20 | 98.40 | 98.40 |
| Mean | 90.76 | 89.96 | 90.50 | 97.62 | 97.70 | 97.71 |
| Std. dev. | 0.82 | 1.42 | 0.95 | 0.61 | 0.75 | 0.61 |

neural networks to automatically learn features of regular textures. Textures form existed databases were classified into regular and irregular; each category consisted of a variety of textures. To a large extent, we evaluated the ability of deep CNNs for modelling the texture regularity. The results obtained can provide useful guidance for concrete analysis of regular textures. The classic CNN model Inception-ResNet-v2 used in a standard way and its more transferable Fisher vector representation both show a high classification result of 98%, which is much higher than 90% obtained by the traditional SIFT features. The trained CNN can correctly recognize most regular patterns, but has lower accuracy on borderline cases, *e.g.* near-regular textures and hybrid textures. Beyond the impressive result, there are still several limitations. First, 2,460 images may be still insufficient to evaluate the methods. More images are needed. In addition, this experiment shows the architectures of current CNNs do not use long-range structures and thus decreases their ability to model complex features such as texture regularity. In the future study we will investigate the way to learn the long range features by CNN models.

## CONCLUSIONS

In this paper we have introduced a new regular texture database. It supplements state-of-the-art texture analysis databases towards perception understanding. Based on the created regular texture databases, we proposed a generalized regular texture modelling and recognition framework. The trained deep CNN models and further generated Fisher representations were robust to different texture layouts, pattern complexity, texton variability and viewing angles, and proved transferable from one domain to another. Our

experiments showed that both methods reached remarkable accuracy, with a best performance of 98% for general regular texture classification.

To further understand and improve the performance of our proposed network, an ablation study has been presented. Although convolutional neural networks take spatial relations into account by pooling local features into a global representation using fully-connected layers, they are known to perform sub-optimally when learning long-range patterns. In our future work, we will focus on applying this generalized regular texture modelling to efficient repetitive elements detection in real-life applications.

### Funding

This work was supported by the National Natural Science Foundation of China (Nos. 61806023 and 61572083), the Henan Provincial Department of Transportation Science and Technology Project (No. 2021G8). The funders had no role in study design, data collection and analysis, decision to publish, or preparation of the manuscript.

### Grant Disclosures

The following grant information was disclosed by the authors:
National Natural Science Foundation of China: 61806023 and 61572083.
Henan Provincial Department of Transportation Science and Technology Project: 2021G8.

### Competing Interests

Xizhi Li is employed by Henan Highway Development Co. LTD. The authors declare that they have no competing interests.

### Author Contributions

- Ni Liu conceived and designed the experiments, performed the experiments, analyzed the data, performed the computation work, prepared figures and/or tables, authored or reviewed drafts of the paper, and approved the final draft.
- Mitchell Rogers performed the experiments, performed the computation work, prepared figures and/or tables, authored or reviewed drafts of the paper, and approved the final draft.
- Hua Cui analyzed the data, authored or reviewed drafts of the paper, and approved the final draft.
- Weiyu Liu performed the experiments, authored or reviewed drafts of the paper, and approved the final draft.
- Xizhi Li performed the computation work, authored or reviewed drafts of the paper, and approved the final draft.
- Patrice Delmas conceived and designed the experiments, authored or reviewed drafts of the paper, and approved the final draft.

## Data Availability

The code files for implementing convolutional neural networks. Fisher vector pooling and SVM solver and the regular textures dataset are available in the Supplemental Files and at GitHub: https://github.com/NiLiu64/Regular-texture-recognition.

## Supplemental Information

Supplemental information for this article can be found online at http://dx.doi.org/10.7717/peerj-cs.869#supplemental-information.

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
