# Peer review of "Deep convolutional neural networks for regular texture recognition"

_PeerJ Computer Science, doi:10.7717/peerj-cs.869_

## Round 0.1 · original submission · Major Revisions

Good effort. More quality of the article some improvement is required based on the reviewers' suggestions.

·

Basic reporting

The paper provides a good and sufficient background of Deep CNN that can classify regular and irregular texture images. Though the approach is relatively simple to implement, yet it is quite powerful.

No significant limitations were discussed. It may be worthwhile to mention the tradeoffs involved in choosing the Deep CNN as opposed to some other technique

Professional English used. No ambiguity

Experimental design

The experimental setup is quite standard, and is appropriate for the study, especially the evaluation and comparison of several different classical CNN models, but is the sample size taken for testing sufficient to represent a large database images

Validity of the findings

I think the motivation for this study need to be made still clearer. In particular, the connection between
(a) the necessary of recognizing and locating regular textures by giving applications, and (b) the necessity of choosing Regular Texture Recognition

Reviewer 2 ·

Basic reporting

- This paper proposed a new texture database that includes 2800 images (1300 regular and 1500 irregular images). The author proposed using five state-of-the-art CNN models; AlexNet, InceptionV3, ResNet50, InceptionResNetV2, and texture CNN (T-CNN), to examine the texture database. The result showed that the InceptionResNetV2 obtained the highest accuracy on the texture database with an accuracy of 97.0%.

- However, this paper did not show any novel method or did not show any complex experiments, such as experiments on insufficient data (10%, 20%, 30% of training set), proposed a new CNN method; Fusion CNNs, 1D-CNN, etc., or show the parameter tuning.

Experimental design

This paper doesn't show any new experiments. It presented only which CNN model performs better in the proposed texture dataset and show only one experiment (see Table 3). The author should do more experiments, at least three experiments. The author should concern about the novel method or exciting experiments.

Validity of the findings

This paper showed only which CNN model performs the best on the proposed database. Then, the author should provide a discussion section on why the InceptionResnetV2 outperformed other CNNs.

Additional comments

- The author should concern more about the novelty of this work. Provide only a new texture database was not sufficient to publish in the outstanding journal.
- The abstract section is relatively short. It did not include any new information about the proposed deep convolutional neural networks (CNNs). However, this paper provides a new regular texture database and uses only standard CNN in the experiments.
- The author should do more experiments, at least three experiments. The author should concern about the novel method or exciting experiments.

Reviewer 3 ·

Basic reporting

There were no significant constraints discussed. It's probably worth mentioning the tradeoffs involved in using deep CNN instead of another approach

Experimental design

This article does not much new experiments over traditional. Only one CNN model performs better in the proposed texture dataset and show only one experiment. It should be compare with more methods see table 3 . The author should do some more experiments, atleast three experiments.

Validity of the findings

Entire article discussed only which CNN model performance the best on the proposed database. Then, the author should summarise why inception ResnetV2 outperformed other CNNs.

Additional comments

The author should concern more about the novelty of this work. Provide only a new texture database not much technical strength to publish in the outstanding journal.
The abstract section is relatively short should be only 300 words
The authors should do more experiments at least three methods.

---

## Round 0.2 · Minor Revisions

A few more points need to be addressed.

·

Basic reporting

The revised manuscript submitted provided a good and sufficient background of Deep CNN that can classify regular and irregular texture images.
The author also mentioned the tradeoffs involved in choosing the Deep CNN as opposed to some other technique

Experimental design

The new manuscript designed provided more complex experiments using 10-fold cross- validation and is sufficient enough to validate the approach

Validity of the findings

(a) the necessary of recognizing and locating regular textures by giving
applications, and
(b) the necessity of choosing Regular Texture Recognition

The above 2 points has now been made clearer in the revised manuscript

Reviewer 2 ·

Basic reporting

In this paper, the author proposed a new texture database collected from Flickr and google images, including 1230 regular textures and 1230 irregular textures. The author proposed to use fisher vector pooling-CNN (FV-CNN) to extract the local feature from the images. In the experimental results, three CNN architectures were evaluated; ResNet-50, InceptionV3, and Inception-ResNetV2. The result showed that the Inception-ResNetV2 outperformed all CNNs with an accuracy of 96% with 10-fold cross-validation. Figure 6, however, could the author shows both train and validation losses and train and validation accuracy. The author showed the corrected classification of regular and irregular textures in Figures 7 and 9. However, could the author show the confusion matrix? So, the audience could see all the correct and incorrect numbers. In addition, the result of the Inception-ResNetV2 and FV-CNNreg-IncepResv2 were 96% and 97% accuracy, respectively. Is it a significant result?

Experimental design

The author provided the experimental results of the FV-CNN. Also, a 10-fold cross-validation method was proposed. The author also compared the FV-CNN with the FV-SIFT method. However, the experimental result is not enough. The author should do more experiments as suggested before.

Validity of the findings

In the new manuscript, the author proposed the FV-CNN to classify the texture images. In the experimental result, the author showed that the FV-CNN outperformed the original CNNs and FV-SIFT method. So, could the author report the kernel of the SVM used in the experiment and report the hyperparameter values. Could the author add the "Fisher Vector" into the framework (Figure 5)? In the framework, it was unclear how the author attached the FV into the CNN?

Additional comments

- I suggest the author do more experiments, such as data augmentation techniques, tuning hyperparameters, etc.
- Would you please provide the Discussion section?

---

## Round 0.3 · Minor Revisions

The authors have done good work and improved the quality of the article. Just a few more final changes are required.

Reviewer 2 ·

Basic reporting

More concerning points are as follows.
- The author responded that the confusion matrix was added as Table 4. But, Table 4 showed the misclassified images by Inception-ResNet-v2.
- The author presented training and validation in Figures 6, 10, and 11. However, the quality of the Figure is relatively low resolution and see some noises. The legend should not overlap on the graph.
- In Figures 7, 8, and 9, what are the numbers under each pattern? The author should describe in the caption or in the text.
- Up-to-date references (2019-2021) are required.

Experimental design

The author clearly explained the experimental results. The experiments are cover the contribution.

Validity of the findings

In Figure 5, please check the block of Fisher vector pooling. Somethings appeared before the word "pooling."

Additional comments

- The author responded by adding the experiments of data augmentation. I found the words "data augmentation features" in line 234. However, I cannot see the compare results. Could the author recheck it?
- The discussions section was presented in this revised manuscript.

---

## Round 0.4 · accepted · Accept

You did good work and addressed all concerns.